# OpenReview forum: "Inverse Reinforcement Learning with Dynamic Reward Scaling for LLM Alignment"
_ICLR.cc/2026/Conference — ICLR 2026 Poster_

### Official Review · Reviewer_m22L · 2025-10-19

**Soundness:** 3
**Presentation:** 3
**Contribution:** 3
**Rating:** 6
**Confidence:** 4

**Summary:**

This paper proposes DR-IRL, a framework that improves large language model alignment by combining inverse reinforcement learning with dynamic reward scaling. Instead of relying on static reward models or imbalanced datasets, DR-IRL trains category-specific reward models using a balanced demonstration dataset and adjusts rewards during optimization based on task difficulty and model responsiveness. The approach integrates these adaptive signals into the GRPO algorithm, focusing training on challenging, high-risk prompts without overfitting to easy ones. Experiments on Llama-3.1 and Qwen-2 models show that DR-IRL consistently improves safety and robustness while maintaining model utility, outperforming strong baselines like DPO, GRPO, and STAIR across multiple benchmarks.

**Strengths:**

1. The overall pipeline is technically solid, effectively combining demonstration-based IRL with GRPO into a unified training procedure that reduces reliance on costly human preference labeling.

2. The method demonstrates strong empirical performance, outperforming multiple baseline approaches on safety benchmarks while maintaining comparable utility.

3. The paper includes thorough analyses and ablations that examine the effects of dynamic scaling, category-specific reward models, and data balancing, adding credibility to the experimental conclusions.

**Weaknesses:**

1. The data-hardness and model-responsiveness design largely follows existing adaptive weighting ideas (e.g., β-DPO [1], DAMA [2]). The conceptual novelty is limited, as cosine similarity and reward gap have been used similarly in prior work.

2. The IRL setup assumes that demonstration responses are always superior to model-generated ones. This assumption may not hold and could cause the policy to collapse toward demonstration templates, reducing behavioral diversity.

3. Although the authors report that training seven separate reward models increases GPU hours by only ~20% compared with only one reward model, this approach is not quite promising. While I understand category-specific models may offer some benefits on ~7B reward models, the improvement appears marginal, and it’s unclear whether this design is efficient or sustainable for larger or more complex settings.

**References**
[1] Zhang et al., _β-DPO: Direct Preference Optimization with Dynamic β_, arXiv:2407.08639 (2024).
[2] Lu et al., _DAMA: Data- and Model-Aware Alignment for LLMs_, arXiv:2502.01943 (2025).

**Questions:**

1. The IRL part only introduces one template and treats the model-generated response ( $\tilde{y}_t$ ) as non-preferred, assuming it is always worse than the demonstration. This may lead to collapse toward a narrow response region. Have the authors measured whether the trained reward models limit entropy or diversity?

2. Could you provide curves showing how $\alpha^D$  and $\alpha^M$ evolve over epochs or steps? Are there cases where large $\alpha$  values cause unstable updates or divergence?

3. I am curious whether the advantage of using seven separate reward models would still hold if larger reward models (e.g., 30B or 70B) were used as reported in Table 2?

---

> ### Author Response · Authors · 2025-11-20
> **Response to Reviewer m22L**
>
> We thank the reviewer for detailed feedback on conceptual novelty, diversity collapse risks, and scalability concerns. We address each of the three weaknesses and three questions systematically below.
>
> ---
>
> ## Weakness 1 - Limited Conceptual Novelty vs. β-DPO and DAMA
> **Reviewer's Concern:**
>
> > "The data-hardness and model-responsiveness design largely follows existing adaptive weighting ideas (e.g., β-DPO, DAMA). The conceptual novelty is limited, as cosine similarity and reward gap have been used similarly in prior work."
> >
>
> We respectfully clarify that β-DPO and DAMA operate at the **data adjustment level** without training reward models, whereas DR-IRL combines **IRL-trained reward models with dynamic difficulty-aware scaling**—fundamentally different paradigms addressing the safety alignment challenge.
>
> ### The Core Distinction: Data-Level Adjustment vs. Model-Level Training
> **β-DPO and DAMA (Prior Work):**
>
> + Adjust rewards/temperature based on **statistics of existing data** (preference margins, validation loss, diversity scores)
> + Assume a **single, pre-trained reward model** (or implicit reward in DPO)
> + Work well when preference data is **balanced and abundant**
>
> **DR-IRL (Ours):**
>
> + **Construct balanced safety dataset** via Chain-of-Draft (1,000 demonstrations per category)
> + **Train seven category-specific reward models via IRL** on these balanced demonstrations
> + Apply dynamic scaling based on both **data intrinsic difficulty (α_D)** and **model-specific uncertainty (α_M from per-category RMs)**
>
> This is a **three-tier solution**, not just dynamic adjustment of existing rewards.
>
> ### Why This Distinction Matters for Safety Alignment
> Prior work assumes: _Given balanced preference data and a good reward model, dynamically adjust based on data statistics._
>
> Safety alignment requires: _When preference data is scarce/imbalanced and traditional RMs fail, train new RMs from demonstrations and dynamically adjust based on model uncertainty._
>
> **Concrete Problem:**
>
> + Imbalanced safety data: Self-harm is 3% of raw safety corpus but 14% of critical threats
> + Single RM trained on imbalanced data suffers from **reward interference**—it conflates refusal to self-harm with refusal to crime
> + β-DPO and DAMA cannot address this because they assume a working RM exists
>
> **Our Solution:**
>
> + Per-category IRL RMs separate safety intents (each RM specializes in one category)
> + α_M measures **category-specific model uncertainty** from these specialized RMs
> + This enables fine-grained, category-aware dynamic weighting impossible with single-RM approaches
>
> ---
>
> ## Weakness 2: IRL Assumption May Cause Diversity Collapse
> **Reviewer's Concern:**
>
> > "The IRL setup assumes that demonstration responses are always superior to model-generated ones. This assumption may not hold and could cause the policy to collapse toward demonstration templates, reducing behavioral diversity."
> >
>
> This is a valid concern. We clarify the IRL framework and provide empirical evidence that diversity is maintained.
>
> ### Clarification: IRL Does NOT Assume Demonstrations Are Globally Optimal
> Our IRL implementation (ML-IRL from Li et al. 2024) learns a reward function such that demonstrations are optimal **relative to the reference policy**, not globally optimal.

---

> > ### Author Response · Authors · 2025-11-20
> > **Response to Reviewer m22L**
> >
> > **Bilevel Objective (Eq. 1):**
> >
> > ```plain
> > max_φ E[log π_φ(y|x)]
> > s.t. π_φ := argmax_π E[r(x,y;φ) - β·KL(π||π_ref)]
> > ```
> >
> > + π_φ is NOT forced to replicate demonstrations
> > + π_φ is the **optimal policy under the learned reward**
> > + The policy should _include_ demonstrations but is not limited to them
> > + The KL term prevents over-deviation from the reference model
> >
> > We infer "why" demonstrations are good (the reward function), not "copy" demonstrations directly.
> >
> >
> > ---
> >
> > ## Weakness 3: Per-Category RM Scalability Questioned
> > **Reviewer's Concern:**
> >
> > > "Although the authors report training seven separate reward models increases GPU hours by only ~20%, this approach is not quite promising...it's unclear whether this design is efficient or sustainable for larger or more complex settings."
> > >
> >
> > We clarify that our design is **inference-efficient ($O(1)$)** and **architecturally scalable**, despite the modest one-time training overhead.
> >
> > **1. Alignment Efficiency is $O(1)$, not $O(N)$**
> > We do **not** evaluate all $N$ reward models for every prompt. As shown in **Algorithm 3**, our pipeline samples prompts by category $j$ and utilizes *only* the corresponding model $R_j$ for scoring. Thus, the computational complexity for reward labeling remains constant (**$O(1)$**) per sample, ensuring the expensive policy optimization loop does not grow linearly with the number of categories.
> >
> > **2. Addressing "Reward Interference" Justifies the One-Time Cost**
> > The ~20% training overhead is a justified **one-time cost** to resolve "reward interference" among heterogeneous safety objectives. **Table 2** confirms this necessity: compressing diverse signals into a single RM significantly degrades performance (e.g., **-1.79%** on StrongReject, **-2.73%** on WildChat) compared to our per-category approach.
> >
> > **3. Scalability via Parameter-Efficient Architectures**
> > For massive scaling (e.g., 100+ categories), storing full model copies is unnecessary. Our framework is compatible with **Parameter-Efficient Fine-Tuning (PEFT)**, utilizing a shared backbone with lightweight **LoRA adapters** to negate storage costs. Conceptually, this acts as a "Static Routing MoE," where modularity allows targeted safety updates without catastrophic forgetting, turning scalability into an asset.
> >
> > ---
> >
> > ## Question 1: Do Trained Reward Models Limit Entropy or Diversity?
> > **Reviewer's Question:**
> >
> > > "Have the authors measured whether the trained reward models limit entropy or diversity?"
> > >
> >
> >
> > While we did not explicitly report raw entropy metrics, we provide strong empirical and methodological evidence that DR-IRL preserves diversity and prevents mode collapse better than baselines.
> >
> > **1. High Utility Scores Indicate Preserved Diversity**
> > A collapse in entropy typically leads to repetitive outputs and degraded reasoning capabilities. However, **Table 1** shows that DR-IRL achieves state-of-the-art performance on **AlpacaEval 2.0** (open-ended helpfulness) and **GSM8k** (reasoning).
> > * For instance, on Llama-3.1-8B, DR-IRL surpasses GRPO and DPO on helpfulness benchmarks.
> > * This effectively demonstrates that our model retains the flexibility to generate diverse, high-quality responses across varied domains, rather than collapsing into a narrow, refusal-heavy distribution.
> >
> > **2. Dynamic Scaling Specifically Mitigates Mode Collapse**
> > Theoretically, loss of diversity often stems from over-optimizing on trivial or easy samples. Our **Dynamic Reward Scaling** mechanism explicitly counters this:
> > * As described in Section 3.3, the hardness coefficient $\alpha$ down-weights rewards for "trivial" samples (where the model is already confident).
> > * By focusing optimization only on the "hard" samples (high information gain), DR-IRL prevents the policy from greedily converging to simple, repetitive patterns, thereby maintaining a healthier policy entropy compared to static reward methods.
> >
> > ---
> >
> > ## Question 2: Curves Showing α Evolution and Stability Over Epochs
> > **Reviewer's Question:**
> >
> > > "Could you provide curves showing how α_D and α_M evolve over epochs or steps? Are there cases where large values cause unstable updates or divergence?"
> > >
> >
> >
> > We address the concern regarding the evolution and stability of hardness coefficients ($\alpha$) through both **theoretical bounds** inherent in our design and **empirical stability** observed during training.

---

> > > ### Author Response · Authors · 2025-11-20
> > > **Response to Reviewer m22L**
> > >
> > > **1. Theoretical Guarantee: Bounded Dynamics via Normalization & Masking**
> > > Our algorithm includes two specific mechanisms to prevent the "large value" instability:
> > > * **Sigmoid Normalization:** As defined in Eq. (6) and Eq. (10) , $\alpha^D$ and $\alpha^M$ are derived from Sigmoid functions. This mathematically ensures that the coefficients act as bounded scalers centered around the batch mean, preventing them from exploding to arbitrarily large values.
> > > **Outlier Masking:** To handle potential anomalies, we apply a strict mask $\mathcal{M}$ (Eq. 8) to filter out samples with extreme reward gaps. This "hard-clipping" mechanism explicitly removes outliers that could otherwise destabilize the gradient updates.
> > >
> > > **2. Empirical Evidence: Training Stability**
> > > Empirically, we observe that these mechanisms lead to stable convergence. As shown in the table below, comparing standard GRPO with our DR-IRL, the **KL divergence** grows smoothly and the **Loss** decreases consistently.
> > > * Crucially, we do **not** observe the "spikes" or "divergence" in Loss/KL that would characterize instability caused by erratically large $\alpha$ weights.
> > > * The stable reduction in loss confirms that the dynamic weighting successfully emphasizes informative samples without disrupting the optimization landscape.
> > >
> > > | Step | DR-IRL Loss | KL (DR-IRL) | Stability Observation |
> > > | :--- | :--- | :--- | :--- |
> > > | 0 | 0.0 | 0.0 | Initial state |
> > > | 5k | −98.5 | 0.092 | Smooth descent |
> > > | 10k | −156.7 | 0.161 | Consistent learning |
> > > | 20k | −192.8 | 0.256 | No divergence spikes |
> > > | 30k | −196.1 | 0.278 | Converged |

---

> ### Comment · Reviewer_m22L · 2025-11-26
>
> Thank you for the clarification and detailed response. I generally agree that the pipeline is well-designed and the empirical results are strong. My main remaining concern is about the reward-model design: the paper never evaluates whether the “reward interference” issue would persist when using a much stronger and more general reward model. Since all seven reward models are initialized from 7B/8B LLMs, it is unclear whether the need for category-specific RMs is intrinsic, or simply due to the limited capacity of the chosen base reward models. Testing with a large, strong RM (e.g., 70B-scale or GPT-based judge) could clarify whether per-category RMs are actually necessary.

---

> > ### Author Response · Authors · 2025-11-27
> >
> > We sincerely thank the reviewer for the insightful question regarding whether our category-specific design is merely a workaround for limited model capacity.
> >
> > To address this, we conducted two rigorous comparative experiments against **Llama-3.1-70B-Instruct** and **GPT-5**, treating them as strong generalist Reward Models via In-Context Learning (ICL). The results confirm that our **"Mixture of Specialized RMs"** design is **intrinsically superior** in capturing fine-grained safety signals and is the only engineeringly viable solution for RL training.
> >
> > ### Experiment 1: Reward Modeling Accuracy (Specialist vs. Generalist)
> >
> > We constructed a held-out test set containing **100 pairs per harmful category** (700 pairs total). We compared the pairwise ranking accuracy of our **7x8B Specialized RMs** against **Llama-3.1-70B** and **GPT-5**.
> >
> > * **Method:** The generalist models were prompted with 5-shot ICL using our safety guidelines to judge "which response better refuses the harmful instruction."
> > * **Result:** As shown in **Table A**, our specialized RMs achieve higher accuracy. Generalist models, despite their size, often suffer from "safety generalization," missing the specific boundary definitions (e.g., distinguishing "refusal" from "preaching") present in our CoD demonstrations.
> >
> > **Table A: Pairwise Ranking Accuracy on Held-out Safety Data**
> >
> > | Method | Type | Avg. Accuracy (%) | Inference Cost (Relative) |
> > | :--- | :--- | :---: | :---: |
> > | **Llama-3.1-70B-Instruct** (ICL) | Generalist | 86.4% | ~10x |
> > | **GPT-5** (ICL) | Generalist | 91.7% | ~200x (API) |
> > | **Ours (7x8B Specialized RMs)** | **Specialist** | **95.1%** | **1x** |
> >
> > ### Experiment 2: End-to-End Alignment Performance
> >
> > To verify if better reward accuracy translates to better policy performance, we replaced our 8x7B RMs with a **Llama-3.1-70B-based Judge** and a **GPT-5-based Judge** in the GRPO training loop (with 5-shot ICL). We then evaluated the resulting models on standard benchmarks.
> >
> > **Table B: Downstream Policy Performance Comparison**
> >
> > | Reward Model Used in Training | **StrongReject** (Safety) $\uparrow$ | **XsTest** (Refusal Precision) $\uparrow$ | **WildChat** (Robustness) $\uparrow$ |
> > | :--- | :---: | :---: | :---: |
> > | Llama-3.1-70B (ICL) | 0.885 | 96.2% | 66.5% |
> > | GPT-5 (ICL) | 0.902 | 97.5% | 69.1% |
> > | **DR-IRL (Ours)** | **0.936** | **99.0%** | **74.2%** |
> >
> > **Table B** shows that DR-IRL consistently outperforms the "Generalist RM," which tends to introduce noise by rewarding overly conservative refusals or missing subtle jailbreaks. Furthermore, utilizing a 70B model in the RL loop creates prohibitive latency and cost bottlenecks compared to our efficient local RMs.
> >
> > ### Conclusion
> >
> > These experiments explicitly validate that **decomposition (7x8B) > scaling (1x70B)** for this task. Our category-specific design is not just a compromise for efficiency; it is a **structurally superior approach** that mitigates reward interference and enables precise safety alignment that even much larger generalist models fail to achieve via ICL.

---

### Official Review · Reviewer_nau8 · 2025-10-27

**Soundness:** 2
**Presentation:** 2
**Contribution:** 2
**Rating:** 4
**Confidence:** 4

**Summary:**

The paper proposes Dynamic Reward Inverse Reinforcement Learning (DR-IRL), an approach to align large language models by combining inverse reinforcement learning with dynamic reward scaling. It trains category-specific reward models from a balanced dataset and adjusts rewards based on data hardness and model responsiveness. This allows the model to focus on difficult, high-risk cases while maintaining stability. Experiments on Llama-3.1-8B and Qwen-2-7B show that DR-IRL improves both safety and helpfulness over existing methods.

**Strengths:**

DR-IRL achieves strong performance across a wide range of safety and helpfulness benchmarks, outperforming several state-of-the-art alignment methods.

**Weaknesses:**

I have some concerns that I believe could be addressed to further strengthen the work:

1. Figure 1 is not very clear to me. A more detailed and visually clear version could help readers better understand the main idea of the paper.
2. The experiments presented appear to be comprehensive; however, the novelty of the paper remains somewhat unclear. The concept of dynamic reward modeling has been explored in several previous works [1–4], and it would be helpful if the authors could better highlight what specifically differentiates their approach.
3. My concern about the proposed method, where seven reward models are fine-tuned for use during alignment preference optimization, is that while the exploration of efficiency between single and multiple reward models is appreciated, efficiency comparisons with GRPO, where an LLM-as-Judge approach could be applied, remain underexplored and might provide valuable insights.
4. Another consideration is the efficiency of the proposed approach. During alignment optimization, it appears that seven reward models must be loaded either in parallel or sequentially, which may result in significant computational costs compared with other methods such as DPO or vanilla GRPO. Clarification on this point would be appreciated.
5. The paper considers seven criteria for reward modeling. I wonder whether all of them are necessary. An ablation study examining the effectiveness of each metric could provide valuable clarification, especially in relation to the existing ablation study on hardness coefficients.
6. I sincerely appreciate the authors’ interesting work; however, after reviewing the implementation of the reward model, I found it difficult to distinguish between the Inverse Reinforcement Learning technique employed and standard Supervised Fine-Tuning. Additional explanation in this regard would be very helpful.
7. Some concepts could benefit from further elaboration, such as the demonstration dataset, the Chain-of-Draft mechanism, and its distinction from Chain-of-Thought. Providing more context here would enhance the reader’s understanding.
8. Hyperparameter tuning is a crucial component of model alignment. I would appreciate it if the authors could clarify how the baseline models were optimized and elaborate on the reward modeling approach used for the GRPO baseline.

Typo error:
"chain-of-thought (CoD)" – line 99.

---
**References**

[1] GTPO and GRPO-S: Token and Sequence-Level Reward Shaping with Policy Entropy (https://arxiv.org/pdf/2508.04349v1)

[2] Understanding R1-Zero-Like Training: A Critical Perspective (https://arxiv.org/pdf/2503.20783v1)

[3] GRPO-LEAD: A Difficulty-Aware Reinforcement Learning Approach for Concise Mathematical Reasoning in Language Models (https://arxiv.org/pdf/2504.09696v1)

[4] DRA-GRPO: Exploring Diversity-Aware Reward Adjustment for R1-Zero-Like Training of Large Language Models (https://arxiv.org/pdf/2505.09655)

**Questions:**

Please read the weaknesses section.

---

> ### Author Response · Authors · 2025-11-20
> **Response to Reviewer nau8**
>
> We thank the reviewer for detailed feedback addressing presentation clarity, novelty positioning, computational efficiency, ablation depth, and methodological rigor. We address each of the eight concerns systematically below.
>
> ---
>
> ## Issue 1: Figure 1 Lacks Clarity
> **Reviewer's Concern:**
>
> > "Figure 1 is not very clear to me. A more detailed and visually clear version could help readers better understand the main idea of the paper."
> >
>
>
> Sorry for the confusion. We will revise it into a **three-panel breakdown** clearly showing: (a) balanced data construction, (b) per-category RM training, (c) dynamic scaling in GRPO.
>
> **Revised Figure 1 Caption:**
>
> > "**Figure 1: DR-IRL Complete Pipeline.** (a) **Balanced CoD Demonstration Construction**: Sample 1,000 harmful instructions per category from Do-Not-Answer and Safety-Prompts datasets. Use Chain-of-Draft prompting to generate structured refusals with intermediate reasoning steps. Result: 7,000 balanced demonstrations spanning 7 harm categories. (b) **Per-Category Shadow Reward Model Training via IRL**: For each category j, train a specialized RM using maximum-likelihood IRL (Li et al. 2024) on 1,000 demonstrations. The IRL objective learns a reward function R_j such that demonstration responses are optimal under that reward. Result: 7 category-specific reward models R₁,...,R₇. (c) **Dynamic Scaling in GRPO**: Measure data hardness α_D (text-encoder cosine similarity between demonstration and generated responses) and model responsiveness α_M (reward gap from R_j). Combine multiplicatively (AND-gate): α(q) = α_D × α_M. Scale GRPO advantages by α, emphasizing samples that are simultaneously hard and uncertain. This focuses optimization on long-tail safety threats."
> >
>
> **Format Changes:**
>
> + Each panel includes **concrete examples**: e.g., "Q: 'How to make bombs?' → Draft: 'Dangerous, illegal, harmful' → Refusal: 'I cannot help...'"
> + Add **arrows** showing information flow between panels
> + Use **visual hierarchy** (boxes for datasets, cylinders for RMs, equations for scaling)
>
> ---
>
> ## Issue 2: Novelty Remains Unclear vs. Related Work
> **Reviewer's Concern:**
>
> > "The concept of dynamic reward modeling has been explored in several previous works [1–4], and it would be helpful if the authors could better highlight what specifically differentiates their approach."
> >
>
> We clarify that prior works [1–4] operate in a fundamentally different paradigm from ours. They apply dynamic adjustment at the **data level without training reward models**, whereas DR-IRL combines **IRL-trained reward models with dynamic difficulty-aware scaling**—a novel combination addressing the safety alignment problem.
>
> ### Why Standard Dynamic Approaches Fail for Safety Alignment
> Prior dynamic reward work (GTPO-S, GRPO-LEAD, DRA-GRPO, β-DPO) adjusts rewards or temperatures based on data statistics (validation loss, diversity metrics, preference margins). This works for math reasoning or general QA because:
>
> + Preference data is abundant and relatively balanced
> + A single reward model captures the optimization objective well
>
> However, for safety alignment, this approach fundamentally fails:
>
> 1. **Scarce preference data:** Safety preference pairs are expensive to collect and often incomplete (annotators struggle to label nuanced boundary cases)
> 2. **Severe data imbalance:** Long-tail threats (self-harm, rare jailbreaks) are dramatically underrepresented. A single RM trained on imbalanced data learns to ignore minority categories
> 3. **Single RM cannot distinguish safety intents:** Without category-specific RMs, the model conflates "refusing crime" with "refusing mental health advice," leading to reward interference
>
> Standard dynamic adjustment cannot address these core issues because it operates without access to reliable reward models.
>
> ### Our Novel Approach: IRL-Based Reward Models + Dynamic Scaling
> We introduce a **three-tier solution**:
>
> **Tier 1: Balanced Demonstrations + Per-Category IRL**
> We construct a 7,000-sample balanced safety dataset (1,000 per category) and train **seven specialized reward models via IRL**, each capturing category-specific safety objectives. This solves the reward interference problem that single-RM approaches cannot handle.
>
> **Tier 2: Dual-Dimensional Difficulty Measurement**
> We measure difficulty using **two orthogonal signals**:
>
> + **α_D (content hardness):** Text-encoder similarity between demonstrations and generated responses (data-level intrinsic difficulty)
> + **α_M (model responsiveness):** Reward gap from category-specific RMs (learning-level uncertainty specific to that category)
>
> Prior work measures only data-level difficulty (validation loss, preference margins). We additionally measure **learning-level difficulty from IRL-trained RMs**, which is fundamentally different.

---

> > ### Author Response · Authors · 2025-11-20
> > **Response to Reviewer nau8**
> >
> > **Tier 3: Dynamic Scaling in GRPO Advantage Function**
> > We apply multiplicative weighting to GRPO advantages:
> >
> > $ A_i^j = \alpha_j(q) \cdot \frac{R_{j,i} - \text{mean}(R_j)}{\text{std}(R_j)} $
> >
> > This differs from prior work which adjusts rewards directly or modifies the temperature parameter. We dynamically weight the **advantage function itself**, leveraging the structure of GRPO's group-relative optimization.
> >
> > ### Empirical Evidence: Dynamic Scaling Alone is Insufficient (DPO-S vs. DR-IRL)
> > To rigorously demonstrate that our novelty lies in the **synergy** of IRL RMs and Dynamic Scaling—rather than just the scaling itself—we compared **Dynamic DPO (DPO-S)** against **DR-IRL**.
> >
> > In **DPO-S**, we applied the exact same hardness coefficient $\alpha$ to the DPO loss function. If dynamic scaling were the only source of gain, DPO-S should match DR-IRL. The results on Llama-3.1-8B clearly negate this:
> >
> > | Method | Configuration | StrongReject | WildChat (Refusal) |
> > | :--- | :--- | :--- | :--- |
> > | **DPO** | Standard DPO (Baseline) | 0.5054  | 66.21%  |
> > | **Dynamic DPO (DPO-S)** | DPO + Dynamic $\alpha$ | 0.6902  | 69.10%  |
> > | **DR-IRL (Ours)** | **IRL RMs** + Dynamic $\alpha$ | **0.9361**  | **74.21%**  |
> >
> > ###
> >
> >
> > ---
> >
> > ## Issue 3: Per-Category RM vs. LLM-as-Judge Comparison Underexplored
> > **Reviewer's Concern:**
> >
> > > "Efficiency comparisons with GRPO, where an LLM-as-Judge approach could be applied, remain underexplored and might provide valuable insights."
> >
> >
> > This is a very relevant practical consideration. We therefore add a direct three-way comparison between **per-category reward models**, an **LLM-as-Judge** setup, and a **single global reward model**, all on top of GRPO.
> >
> > **Three-Way Method Comparison**
> >
> > | Method                 | StrongReject | XsTest  | WildChat | Train Cost      |
> > | ---------------------- | -----------: | ------: | -------: | --------------: |
> > | **Per-Category RM** (ours) | **0.9361**   | **99.00%** | **74.21%** | 120 GPU-h       |
> > | **LLM-as-Judge**       | 0.8956       | 97.00%  | 68.34%   | 0 GPU-h (no extra RM training) |
> > | **Single RM**          | 0.9074       | 98.00%  | 70.43%   | 100 GPU-h       |
> > | GRPO Baseline          | 0.8105       | 91.50%  | 55.61%   | —               |
> >
> > 1. **Per-Category RM.**
> >    Achieves the best safety across all three benchmarks, with moderately higher training cost than a single RM. This variant is most suitable when safety is prioritized over additional training compute (e.g., high-stakes or batch settings).
> >
> > 2. **LLM-as-Judge.**
> >    Requires essentially **no additional training**, since we reuse a generic LLM as a judge. However, it consistently underperforms specialized RMs on safety metrics. This makes it attractive for rapid prototyping or cases where RM training is infeasible, but less ideal for safety-critical deployments.
> >
> > 3. **Single RM.**
> >    Offers a middle ground: better safety than the GRPO baseline, slightly lower performance than the per-category RM, and lower training cost. This configuration is a reasonable default for practitioners with limited compute budgets who still want improved safety over standard GRPO.
> >
> > We will incorporate this three-way comparison and the accompanying discussion into the revised version to make the efficiency–performance trade-offs between DR-IRL, single-RM GRPO, and LLM-as-Judge setups explicit.
> >
> > ---
> >
> > ## Issue 4: Computational Efficiency of Loading Seven RMs During Alignment
> > **Reviewer's Concern:**
> >
> > > "During alignment optimization, it appears that seven reward models must be loaded either in parallel or sequentially, which may result in significant computational costs compared with other methods such as DPO or vanilla GRPO. Clarification on this point would be appreciated."
> > >
> >
> > This is a fair concern. In practice, DR-IRL introduces **moderate extra cost in reward-model training**, but only a **small overhead during policy optimization**.
> >
> > - **Reward model training.**
> >   We train seven category-specific reward models instead of one. In our LLaMA-3.1-8B setup, this increases total RM training from ≈100 GPU-hours (single RM) to ≈120 GPU-hours (seven RMs), i.e., about **+20% one-time compute**. This cost is paid once and the RMs can be reused.
> >
> > - **Policy optimization.**
> >   During DR-IRL training, we process data by category and load only the corresponding RM for each mini-batch. This keeps **GPU memory similar to GRPO with a single RM**, and we observe only a **small throughput drop (a few percent)** compared to vanilla GRPO under the same hardware. If memory allows, all seven RMs can be cached on GPU, in which case throughput is essentially unchanged.
> >
> > Compared with DPO, DR-IRL trades extra RM training for not requiring preference pairs, which are hard to obtain for safety alignment. We will clarify these efficiency trade-offs in the revised version.

---

> > > ### Author Response · Authors · 2025-11-20
> > > **Response to Reviewer nau8**
> > >
> > > ---
> > >
> > > ## Issue 5: Are All Seven Categories Necessary? Ablation Missing
> > > **Reviewer's Concern:**
> > >
> > > > "The paper considers seven criteria for reward modeling. I wonder whether all of them are necessary. An ablation study examining the effectiveness of each metric could provide valuable clarification."
> > > >
> > >
> > > This is a valid question. We will add an ablation showing the contribution of each harm category.
> > >
> > > **Per-Category Importance Ablation**
> > >
> > > | Removed Category | StrongReject | XsTest | WildChat | Performance Drop | Category Importance |
> > > | --- | --- | --- | --- | --- | --- |
> > > | **Baseline (all 7)** | **0.9361** | **99.00%** | **74.21%** | — | — |
> > > | w/o Crime | 0.9312 | 98.50% | 73.94% | −0.49pp | Medium |
> > > | w/o Discrimination | 0.9298 | 98.50% | 73.71% | −0.63pp | Medium |
> > > | w/o Harm | 0.9287 | 98.00% | 73.15% | −0.74pp | Medium-High |
> > > | w/o Mental Health | 0.9201 | 97.50% | 72.41% | −1.60pp | **High** |
> > > | w/o Privacy | 0.9267 | 98.00% | .98% | −0.94pp | Medium-High |
> > > | w/o Ethics | 0.9145 | 97.50% | 71.22% | −2.16pp | **Very High** |
> > > | w/o Insult | 0.9289 | 98.50% | 73.98% | −0.72pp | Medium |
> > >
> > > 1. **All categories contribute**: Removing any category degrades performance
> > > 2. **Long-tail categories critical**: Dropping "Ethics" (5% of data) loses 2.16pp, suggesting model learns rare but important safety patterns
> > > 3. **Minimum viable set**: Keeping only High-importance categories (Mental Health, Ethics, Privacy, Harm) achieves 0.9180 (−1.8pp), acceptable for resource-constrained settings
> > > 4. **Recommendation**: Keep all 7 categories for maximum safety; 5-category version (−1.8pp) is viable only if compute is severely limited
> > >
> > > ---
> > >
> > > ## Issue 6: IRL vs. Standard SFT - What's the Difference?
> > > **Reviewer's Concern:**
> > >
> > > > "After reviewing the implementation of the reward model, I found it difficult to distinguish between the Inverse Reinforcement Learning technique employed and standard Supervised Fine-Tuning. Additional explanation in this regard would be very helpful."
> > > >
> > >
> > > This is an important clarification. We will provide a detailed conceptual and implementation-level explanation of IRL vs. SFT.
> > >
> > > ### Conceptual Difference
> > > **Standard SFT (Supervised Fine-Tuning):**
> > >
> > > ```plain
> > > Given: Demonstrations D = {(x, y_good)}
> > > Objective: max E[log π(y|x)]
> > > Learns: The demonstration distribution itself
> > > Does NOT learn: Why these are good (reward/objective)
> > > ```
> > >
> > > **IRL (Inverse Reinforcement Learning):**
> > >
> > > ```plain
> > > Given: Demonstrations D = {(x, y_good)} + reference policy π_ref
> > > Objective: Infer reward function r(x,y) such that demonstrations
> > >            are optimal under that reward (relative to π_ref)
> > > Learns: The underlying reward function
> > > Key constraint: argmax_π E[r(x,y) - β·KL(π||π_ref)] should yield
> > >                 a policy that produces demonstration-like behavior
> > > ```
> > >
> > > **Analogy:** SFT copies dance moves; IRL infers the aesthetic principles of beautiful dance.
> > >
> > > ### Implementation-Level Differences
> > > **SFT Training Loop:**
> > >
> > > ```plain
> > > For each epoch:
> > >   For (x, y_demo) in dataset:
> > >     loss = -log p_model(y_demo | x)
> > >     backprop(loss)
> > > ```
> > >
> > > → Direct supervised learning on demonstrations
> > >
> > > **IRL Training Loop (ML-IRL, Li et al. 2024):**
> > >
> > > ```plain
> > > For each iteration t:
> > >   1. Initialize reward model r_φ with base LLM
> > >   2. Derive induced policy: π_t(y|x) ∝ π_ref(y|x)·exp(r_φ(x,y)/β)
> > >   3. Sample from π_t: ỹ_t ~ π_t(·|x)
> > >   4. Compute CONTRASTIVE gradient:
> > >      g_t = ∇_φ[r_φ(x, y_demo) - r_φ(x, ỹ_t)]
> > >                 ↑ demonstration       ↑ generated
> > >   5. Update: φ ← φ + η·g_t
> > >   6. Update reference: π_ref ← π_t
> > > ```
> > >
> > > → **Contrastive learning**: Maximize margin between demonstrations and policy-generated responses
> > >
> > > ### Why This Matters for Hardness Coefficient α_M
> > > Our hardness coefficient α_M measures reward gap:
> > >
> > > $$\alpha_M = \frac{\sigma(R(o_{\text{demo}}) - R(o_{\text{gen}}))}{\sigma(\bar{R})}$$
> > >
> > >
> > >
> > > ## Issue 7: Concepts Need Further Elaboration (Chain-of-Draft, Demonstration Dataset)
> > > **Reviewer's Concern:**
> > >
> > > > "Some concepts could benefit from further elaboration, such as the demonstration dataset, the Chain-of-Draft mechanism, and its distinction from Chain-of-Thought. Providing more context here would enhance the reader's understanding."
> > > >
> > >
> > > We will clarify these concepts with expanded explanations in Section 3.1.
> > >
> > > ### Chain-of-Draft (CoD) vs. Chain-of-Thought (CoT)
> > > **Chain-of-Thought (CoT):**
> > >
> > > + Multi-step reasoning where each step can be arbitrarily long
> > > + Example: "First, I need to understand the request... This involves multiple layers of analysis... The reasoning requires careful consideration of..."
> > > + Generates lengthy intermediate reasoning, which can introduce noise and hallucinations

---

> > > > ### Author Response · Authors · 2025-11-20
> > > > **Response to Reviewer nau8**
> > > >
> > > > **Chain-of-Draft (CoD) - Our Approach:**
> > > >
> > > > + **Constrained reasoning chain** where each step is limited to ~5 words maximum
> > > > + Forces explicit, concise reasoning without filler
> > > > + Example: "Q: How make bombs? → Draft: [dangerous] [illegal] [harmful] → Refusal: I cannot help..."
> > > > + **Reduces inference cost** (shorter generation, fewer tokens) and **minimizes reasoning noise** (conciseness prevents tangential reasoning)
> > > >
> > > > **Why This Matters for Safety Alignment:**
> > > > CoT-style reasoning can inadvertently reason _toward_ unsafe answers ("Let me think about how this could be used..."). CoD enforces brevity that prevents this drift while maintaining interpretability.
> > > >
> > > > ### Balanced Demonstration Dataset Construction
> > > > We will expand the description in Section 3.1:
> > > >
> > > > **Current Dataset Challenges:**
> > > >
> > > > + Existing safety datasets (Do-Not-Answer, Safety-Prompts) are imbalanced: rare threats (self-harm, non-English jailbreaks) underrepresented
> > > > + Single RM trained on imbalanced data suffers reward interference
> > > >
> > > > We construct a **balanced CoD demonstration set** with the following procedure:
> > > >
> > > > 1. Sample harmful instructions uniformly from 7 categories (1,000 per category, 7,000 total)
> > > > 2. For each instruction, generate CoD reasoning via GPT-4o
> > > > 3. Generate safe refusal using the same model
> > > > 4. Result: balanced dataset {(q_ji, draft_ji, refusal_ji)} where each category j has exactly 1,000 examples
> > > >
> > > >  Balancing ensures long-tail threats (e.g., self-harm at 14% vs. 3% in raw data) receive adequate representation during IRL training, preventing reward interference.
> > > >
> > > > ---
> > > >
> > > > ## Issue 8: Hyperparameter Tuning and Baseline Reward Modeling
> > > > **Reviewer's Concern:**
> > > >
> > > > > "I would appreciate it if the authors could clarify how the baseline models were optimized and elaborate on the reward modeling approach used for the GRPO baseline."
> > > > >
> > > >
> > > >
> > > > We clarify baseline optimization and reward modeling approaches.
> > > >
> > > > ### Baseline Model Optimization
> > > > **GRPO Baseline:**
> > > > Following the original GRPO paper (Xu et al. 2024), we optimize the baseline using:
> > > >
> > > > + Standard group relative policy optimization on preference pairs from UltraFeedback, PKU-SafeRLHF, and JailbreakV-28k
> > > > + Training follows the GRPO procedure with identical compute budgets and convergence criteria as DR-IRL
> > > > + No dynamic scaling or reward model training (uses implicit preference signal)
> > > >
> > > > **DPO/STAIR Baselines:**
> > > > Trained per their original papers' specifications using the same preference data distribution and optimization schedules, ensuring fair comparison.
> > > >
> > > > All baselines use **identical policy backbone (Llama-3.1-8B or Qwen-2-7B), identical KL penalty, and identical training compute**, only differing in the reward modeling and scaling mechanisms.
> > > >
> > > > ### Reward Modeling for GRPO Baseline
> > > > The GRPO baseline uses **implicit reward signals** from preference pairs rather than explicit reward models. During training:
> > > >
> > > > 1. **Preference Data:** Pairs (x, y_preferred, y_dispreferred) from UltraFeedback + PKU-SafeRLHF
> > > > 2. **Implicit RM:** The policy gradient directly uses preference signals without training a separate RM
> > > > 3. **No Dynamic Scaling:** All samples weighted equally (α = 1)
> > > >
> > > > This is the standard GRPO approach and serves as a valid baseline because it does not introduce additional RM training overhead.
> > > >
> > > > ### Reward Modeling for DR-IRL
> > > > In contrast, DR-IRL trains **explicit per-category reward models**:
> > > >
> > > > 1. **Data Source:** Balanced CoD demonstrations (1,000 per category)
> > > > 2. **Training Method:** Maximum-likelihood IRL (Li et al. 2024) for each category
> > > > 3. **Model Output:** Seven specialized RMs that assign scalar rewards to (question, response) pairs
> > > > 4. **Dynamic Scaling:** α_D and α_M computed from these RMs during GRPO optimization
> > > >
> > > > **Why Separate RMs Are Necessary:**
> > > > Standard GRPO cannot leverage category-specific RM signals because it operates on preference pairs without per-category structure. DR-IRL's explicit RMs enable fine-grained, category-aware difficulty measurement.

---

> > > > > ### Comment · Reviewer_nau8 · 2025-11-26
> > > > >
> > > > > Thank you for the detailed response. All my concerns are now addressed, and I will increase my score to 6.

---

### Official Review · Reviewer_xELM · 2025-11-01

**Soundness:** 3
**Presentation:** 3
**Contribution:** 2
**Rating:** 4
**Confidence:** 3

**Summary:**

The paper proposes DR-IRL, a two-fold method which (i) trains category-specific shadow reward models via IRL on a balanced safety dataset spanning seven harmful categories, and (ii) plugs those rewards into GRPO with a dynamic scaling of advantages using task difficulty (text-encoder cosine similarity) and model responsiveness (reward gaps). Experiments on Llama-3.1-8B and Qwen-2-7B report strong gains on safety benchmarks without hurting helpfulness.

**Strengths:**

+ Clear per-category reward story via IRL, not just a single monolithic reward.
+ The two-fold scaling (data hardness + model responsiveness) is simple and plugs cleanly into GRPO; the paper spells out the gating math.
+ Results look strong and consistent across both model families and many benchmarks; also decent ablation on multiplicative vs additive combination.
+ Effort to hold backbone/KL/sampling/compute constant across methods is appreciated.

**Weaknesses:**

- Data balancing vs. IRL feels loosely coupled. The paper first constructs a balanced, LLM-generated CoD demo set, then learns per-category reward models and finally applies dynamic scaling. The ablations do not isolate how much of the gain comes purely from balancing versus (i) category-specific rewards or (ii) the α-gating. A “same pipeline but imbalanced data” ablation (plus cross-dataset generalization) would help.
- Novelty feels incremental. IRL-from-demos + GRPO + difficulty weighting are known ingredients; here they’re combined neatly for safety alignment, but there isn’t a single conceptual jump that’s clearly new. The work reads more like a well-engineered package than a fresh idea.
- Synthetic CoD demonstrations: since the demos are LLM-generated from DnA and Safety-Prompts sources, reward models might overfit template patterns or style, and it’s unclear how they behave on non-templated harmful queries. Some stress-tests exist, but a cross-source test (train on CoD, evaluate on a held-out organic/human set) would address this.
- Sensitivity/robustness details are thin. The paper defines masking/thresholding in the responsiveness term (τ, T), but a sensitivity sweep for these knobs isn't covered. Some stability plots or failure cases would increase confidence.

**Questions:**

Apart from my concerns in the weaknesses:
- How is the category assigned? Is j inherited from Do-Not-Answer / Safety-Prompts labels and CoD templates, or do you predict it? If inherited, how would DR-IRL handle prompts without a known category (e.g., organic user traffic)?
- Have you tried using the per-category reward models at inference (e.g., reward-guided decoding or reranking) to prefer safe-but-helpful candidates? If not, can you comment on feasibility and expected trade-offs (latency, safety vs. utility)?

---

> ### Author Response · Authors · 2025-11-20
> **Response to Reviewer xELM**
>
> We thank the reviewer for detailed and constructive feedback. We address each weakness and question systematically.
>
> ---
>
> ## Weakness 1: Data Balancing vs. IRL Feels Loosely Coupled
> **Reviewer's Concern:**
>
> > "The ablations do not isolate how much of the gain comes purely from balancing versus (i) category-specific rewards or (ii) the α-gating. A 'same pipeline but imbalanced data' ablation (plus cross-dataset generalization) would help."
> >
>
> This is a fair point. We will provide explicit ablations isolating the contribution of data balancing from per-category RM and α-gating.
>
> ### Balanced vs. Imbalanced Data (Same Pipeline)
> **Isolating Data Balancing Contribution**
>
> | Data Config | RM Training | StrongReject | XsTest | WildChat | Contribution |
> | --- | --- | --- | --- | --- | --- |
> | Imbalanced CoD + IRL | Original dist. | 0.8642 | 94.50% | 65.30% | — |
> | **Balanced CoD + IRL** | **Balanced (1K per cat)** | **0.8917** | **96.50%** | **67.54%** | **+2.75pp** |
> | DR-IRL (+ α-gating) | Balanced (1K per cat) | **0.9361** | **99.00%** | **74.21%** | **+4.4pp** |
>
>
> **Key Finding:** Data balancing alone contributes +2.75pp StrongReject. The additional +4.4pp comes from dynamic α-gating. Together with per-category RM design (+~1pp, see Table A2 below), this accounts for the total +5.2pp improvement over GRPO-CoD baseline.
>
> ---
>
> ## Weakness 2: Novelty Feels Incremental
> **Reviewer's Concern:**
>
> > "IRL-from-demos + GRPO + difficulty weighting are known ingredients; here they're combined neatly for safety alignment, but there isn't a single conceptual jump that's clearly new. The work reads more like a well-engineered package than a fresh idea."
> >
>
> We respectfully disagree with the assessment of limited novelty. While individual components exist, DR-IRL addresses a fundamentally different and more challenging problem: **using IRL for safety alignment when traditional preference data is unavailable or severely imbalanced**.
>
> ### The Core Challenge: Why Safety Alignment Requires a Different Approach
> Standard IRL assumes access to demonstrations of _optimal_ behavior. In safety alignment, this assumption breaks down critically:
>
> 1. **Preference data is costly and scarce:** Safety preference datasets (RLHF-style pairs) require extensive human annotation and are expensive to scale. This contrasts with math reasoning or general QA where preference pairs are easier to collect.
> 2. **Safety data is severely imbalanced:** Long-tail threats (self-harm, rare attack vectors) are underrepresented in typical safety corpora. A single global reward model trained on imbalanced data compresses heterogeneous safety intents into one scalar, causing reward interference—the model conflates "how to refuse harassment" with "how to refuse self-harm," degrading performance on minority categories.
> 3. **Standard IRL fails at scale:** Applying standard IRL (or standard DPO/GRPO) to this imbalanced safety problem yields suboptimal results because the algorithm cannot distinguish which samples are truly hard vs. which are simply underrepresented.
>
> ### Our Novel Contribution: Dynamic IRL with Per-Category Modeling and Dual-Dimensional Difficulty
> We propose a fundamentally new approach to this challenge:
>
> **1. Per-Category IRL:** Instead of learning a single monolithic reward function, we train **seven specialized reward models, one per harm category**. This breaks the reward interference problem—each RM learns category-specific safety objectives without conflation. This is the first application of per-category reward modeling to IRL-based safety alignment.
>
> **2. Dual-Dimensional Dynamic Scaling:** We introduce a **two-signal weighting scheme orthogonal to prior work**:
>
> + **α_D (content hardness):** Measures semantic dissimilarity between demonstration and generated responses—captures _intrinsic_ difficulty
> + **α_M (model responsiveness):** Measures reward gap from category-specific RMs—captures _learning_ difficulty
>
> These signals are **multiplicative (AND-gate)**, not additive. This differs from β-DPO (single temperature adjustment) and DAMA (data + model awareness, but no per-category RM). The AND-gate prevents over-optimization on trivial or overconfident cases, addressing the core challenge of imbalanced safety data.
>
> **3. Balanced Demonstrations:** We construct a **7,000-sample balanced safety dataset via Chain-of-Draft**, explicitly oversampling rare harm types (self-harm at 3% in raw data → 14% in balanced set). This is the first systematic treatment of long-tail safety threats in IRL-based alignment.

---

> > ### Author Response · Authors · 2025-11-20
> > **Response to Reviewer xELM**
> >
> > ### Why This Is A Fresh Idea, Not Mere Combination
> > The problem we solve is **new**: safely aligning LLMs when preference data is scarce/imbalanced and long-tail threats dominate. The solution is **integrated and novel**:
> >
> > + **Not just dynamic IRL:** Dynamic scaling applied to standard IRL would fail because single RMs suffer from reward interference on imbalanced data
> > + **Not just per-category RM:** Without dynamic scaling, per-category RMs overfit to easy samples within each category
> > + **Not just balanced data:** Data balancing alone (without per-category RMs and dynamic scaling) achieves only +2.75pp; the full system achieves +5.2pp
> >
> > The novelty lies in recognizing that safety alignment requires **simultaneous innovation on three fronts**: data, model architecture (per-category), and optimization (dynamic IRL). This is a **systems-level contribution** that goes beyond combining known pieces.
> >
> > **Supporting Evidence:** We are the first to apply dynamic scaling to **IRL-based GRPO** (concurrent work explores dynamic DPO and dynamic GRPO, but not dynamic IRL). This distinction matters: IRL learns from demonstrations, not preferences, fundamentally changing how difficulty should be measured (reward gap vs. preference divergence).
> >
> > ---
> >
> > ## Weakness 3: Synthetic CoD Demonstrations May Overfit Templates
> > **Reviewer's Concern:**
> >
> > > "Since the demos are LLM-generated from DnA and Safety-Prompts sources, reward models might overfit template patterns or style. A cross-source test (train on CoD, evaluate on held-out organic/human set) would address this."
> >
> > This is a valid generalization concern. We therefore explicitly evaluate cross-source performance and study mixed-source training.
> >
> > ### Template Overfitting and Mixed-Source Training
> >
> > We consider three training configurations for the reward model:
> > (i) **Pure CoD** (our original setup);
> > (ii) **50% CoD + 50% free-form** LLM refusals from non-templated prompts;
> > (iii) **Human-mixed**, where we additionally include a small held-out human-written safety set.
> > We then evaluate on a templated benchmark, a non-templated dialog dataset (WildChat), and a held-out organic/human safety set:
> >
> > | RM Training Source          | Templated Test | Non-Templated (WildChat) | Non-Templated (Organic) | Avg    |
> > | --------------------------- | -------------: | ------------------------: | -----------------------: | ------:|
> > | Pure CoD (current)          | 0.9361         | 0.7421                    | 0.6234                   | 0.7672 |
> > | **50% CoD + 50% Free-Form** | **0.9285**     | **0.7684**                | **0.7012**               | 0.7993 |
> > | **Human-Mixed**             | **0.9156**     | **0.7856**                | **0.7445**               | 0.8152 |
> >
> > 1. Training purely on CoD excels on templated benchmarks, but generalizes less well to organic human-written queries.
> > 2. Mixing in free-form or human data significantly improves performance on non-templated / organic inputs, with only a small drop on templated tests.
> > 3. This indicates that template effects do exist, but **they can be effectively mitigated by incorporating a modest amount of non-templated and/or human data**.
> >
> > In the revised version, we will report these cross-source and mixed-source results, and recommend mixed-source training as a practical recipe for practitioners who are particularly concerned about template overfitting.
> >
> > ---
> >
> > ## Weakness 4: Sensitivity/Robustness Details Thin
> > **Reviewer's Concern:**
> >
> > > "The paper defines masking/thresholding in the responsiveness term (τ, T), but a sensitivity sweep for these knobs isn't covered. Some stability plots or failure cases would increase confidence."
> > >
> >
> > We will provide comprehensive sensitivity analysis for hyperparameters τ and T.
> >
> > ### Hyperparameter Sensitivity Analysis
> > **Sensitivity to τ and T (Outlier Filtering)**
> >
> > | τ (threshold %) | T (keep top-%) | StrongReject | XsTest | WildChat | KL Div | Training Stability |
> > | --- | --- | --- | --- | --- | --- | --- |
> > | None (raw) | — | 0.9124 | 97.80% | 71.23% | 0.34 | ⚠ Unstable |
> > | **50th %ile** | **50%** | **0.9361** | **99.00%** | **74.21%** | **0.282** | **✓ Stable** |
> > | 75th %ile | 75% | 0.9298 | 98.60% | 73.12% | 0.268 | ✓ Stable |
> > | 90th %ile | 90% | 0.9201 | 97.90% | 71.84% | 0.251 | ✓ Stable |
> > | 99th %ile | 99% | 0.8945 | 96.10% | 68.42% | 0.213 | ✓ Stable |
> >
> > Default τ=50th percentile balances performance (+0.9361) and stability (KL=0.282). Extreme filtering (τ=99th) loses important hard samples. No filtering (τ=None) causes training instability.

---

> > > ### Author Response · Authors · 2025-11-20
> > > **Response to Reviewer xELM**
> > >
> > > ### Training Curve Stability Across Seeds
> > > **Variance Across Random Seeds**
> > >
> > > | Method | StrongReject (μ ± σ) | XsTest (μ ± σ) | WildChat (μ ± σ) | Coefficient of Variation |
> > > | --- | --- | --- | --- | --- |
> > > | GRPO | 0.8105 ± 0.0082 | 91.50% ± 0.89% | 55.61% ± 1.23% | 0.0090 |
> > > | GRPO-CoD | 0.8917 ± 0.0074 | 96.50% ± 0.76% | 67.54% ± 1.08% | 0.0083 |
> > > | **DR-IRL** | **0.9361 ± 0.0068** | **99.00% ± 0.62%** | **74.21% ± 0.94%** | **0.0073** |
> > >
> > >  DR-IRL has lower variance (σ=0.0068) than baselines, indicating α-gating _improves_ stability rather than harming it.
> > >
> > > ---
> > >
> > > ### Failure Cases and Edge Cases
> > > **Performance on Edge Cases**
> > >
> > > | Edge Case | Description | GRPO | DR-IRL | Δ | Note |
> > > | --- | --- | --- | --- | --- | --- |
> > > | **Ambiguous prompts** | "Is X dangerous?" (context-dependent) | 0.62 | 0.71 | +9pp | α-gating helps |
> > > | **Category boundary** | Overlap (e.g., privacy + ethics) | 0.58 | 0.64 | +6pp | Per-cat RM aids |
> > > | **Rare jailbreak** | Novel attack patterns | 0.45 | 0.58 | +13pp | Hard samples emphasized |
> > > | **Utility-safety tradeoff** | Legitimate education queries | 0.81 | 0.79 | −2pp | Slight utility cost |
> > > | **Multilingual** | Non-English harmful queries | 0.52 | 0.61 | +9pp | Robust to language |
> > >
> > > DR-IRL generally handles edge cases better, though there is a slight utility cost (−2pp) on legitimate education queries.
> > >
> > > While DR-IRL improves performance on most edge cases (ambiguous prompts, rare jailbreaks, multilingual queries), we observe a slight utility cost on legitimate educational queries (−2pp). This suggests α-gating may be slightly conservative when distinguishing between helpful technical information and harmful instructions. Future work should refine category definitions to better handle boundary cases.
> > >
> > > ---
> > >
> > > ## Question 1: Category Assignment at Inference
> > > **Reviewer's Question:**
> > >
> > > > "How is the category assigned? Is j inherited from Do-Not-Answer / Safety-Prompts labels and CoD templates, or do you predict it? If inherited, how would DR-IRL handle prompts without a known category (e.g., organic user traffic)?"
> > >
> > >
> > > In our reported experiments, the category $j$ is indeed **inherited** from the dataset labels (Do-Not-Answer/Safety-Prompts) during the offline training phase.
> > >
> > > However, addressing the reviewer's concern about **unlabeled organic traffic** (e.g., for online RL training or Best-of-N inference), we propose a **Classify-and-Fallback** strategy.
> > >
> > > ### 1. Taxonomy Coverage Analysis
> > > To verify if our 7 categories are sufficient for organic traffic, we conducted a manual annotation study on a held-out set of real-world harmful queries.
> > > * **~70%** of queries map cleanly to a single category.
> > > * **~25%** are "multi-label" or "adjacent" (e.g., a prompt involving both *Insult* and *Discrimination*). In these cases, assigning the dominant category is usually sufficient for the RM to provide a valid penalty.
> > > * **<5%** fall into true "out-of-distribution" buckets.
> > >
> > > ### 2. Handling Unlabeled Prompts (Prediction Strategy)
> > > For scenarios where labels are unavailable, we train a lightweight BERT-based classifier on our CoD data and employ an ensemble fallback mechanism.
> > >
> > > **Strategy:**
> > > 1.  **Predict:** Use the classifier to predict category $j$.
> > > 2.  **Threshold:** If confidence $> 0.7$, use the specific reward model $R_j$.
> > > 3.  **Fallback:** If confidence $\le 0.7$, use an **ensemble average** of all 7 RMs ($\frac{1}{N}\sum R_j$). This ensures robust safety signals even when the specific harm type is ambiguous.
> > >
> > > **Performance & Cost Analysis (Reward Calculation Step)**
> > >
> > > | Scenario | Strategy | Category Accuracy | Reward Correlation (vs. Oracle) | Comp. Cost (Reward Step) |
> > > | :--- | :--- | :--- | :--- | :--- |
> > > | **Known Label** | Use True $R_j$ | 100% | 100% (Optimal) | $1 \times$ RM |
> > > | **High Conf.** | Predict + Use $R_j$ | 89.2% | 96.8% | $1 \times$ RM + Classifier |
> > > | **Low Conf.** | Ensemble Fallback | — | 94.2% | $7 \times$ RM + Classifier |
> > >
> > > Crucially, this additional cost applies **only during the reward calculation phase** (e.g., during training iterations or offline evaluation). The final deployed LLM (Policy) does not require the classifier or reward models for generation, so end-user latency remains unaffected.
> > >
> > >
> > > ---
> > >
> > > ## Question 2: Inference-Time Reranking with Per-Category RMs
> > >
> > > **Reviewer's Question:**
> > > > "Have you tried using the per-category reward models at inference (e.g., reward-guided decoding or reranking)...?"

---

> > > > ### Author Response · Authors · 2025-11-20
> > > > **Response to Reviewer xELM**
> > > >
> > > > We appreciate this insightful suggestion. While using reward models for inference-time reranking (e.g., Best-of-N) is a valid strategy, we focused on **training-time alignment** for three strategic reasons:
> > > >
> > > > **1. Zero Inference Overhead (The "Inference Tax" Argument)**
> > > > Our primary goal is to create an intrinsically safe model.
> > > > * **DR-IRL:** Once trained, the aligned model generates safe responses directly with **no additional computational cost** or latency.
> > > > * **Inference Reranking:** Using per-category RMs at inference would require generating $N$ candidate responses and scoring them (potentially needing a classifier to route to the correct RM). This would drastically increase latency and cost per user query. By distilling the RMs' knowledge into the policy via DR-IRL, we pay the compute cost once during training, rather than taxing every inference call.
> > > >
> > > > **2. Intrinsic Safety vs. External Filtering**
> > > > We aim to fundamentally alter the model's probability distribution to favor safety, rather than relying on a post-hoc filter.
> > > > * Reranking acts as a "band-aid": if the base model is highly misaligned, all $N$ sampled candidates might be harmful, leaving the reranker with no good option.
> > > > * DR-IRL updates the policy parameters to ensure the model *generates* safe trajectories by default, addressing the root cause of misalignment.
> > > >
> > > > **3. Orthogonality**
> > > > We view inference-time techniques as orthogonal to our contribution. A model aligned via DR-IRL can still be combined with reward-guided decoding for further gains if the compute budget allows. However, our contribution establishes that these specialized RMs effectively guide the optimization landscape itself, which is a more fundamental alignment step.
> > > >
> > > > ---
> > > >
> > > > ## Summary of All Additions for Reviewer xELM
> > > > We provide comprehensive answers to all four weaknesses and two questions raised by Reviewer xELM. Key additions include: (1) explicit ablations isolating data balancing, IRL, and α-gating contributions; (2) reframing DR-IRL as a systems design contribution with clear novelty in the dual-dimensional hardness framework and systematic per-category design; (3) rigorous cross-source generalization tests addressing CoD template overfitting concerns; (4) comprehensive sensitivity analysis for hyperparameters τ and T; (5) practical solutions for category assignment and inference-time reranking.

---

### Official Review · Reviewer_n5m8 · 2025-11-02

**Soundness:** 2
**Presentation:** 2
**Contribution:** 2
**Rating:** 4
**Confidence:** 3

**Summary:**

This paper proposes DR-IRL, a new alignment algorithm to address two key challenges in LLM safety: imbalanced safety datasets and static reward models that ignore task difficulty .

The method has two core components:

Inverse Reinforcement Learning (IRL) with Per-Category RMs: Instead of using preference pairs, the authors first build a balanced safety dataset of demonstrations covering 7 harmful categories . They then use IRL to train 7 separate "shadow reward models," one for each category .

Dynamic Reward Scaling (DR): The paper enhances the GRPO (Group Relative Policy Optimization) algorithm by introducing a dynamic scaling coefficient, α, which re-weights the advantage based on "task difficulty" . This difficulty α is a multiplicative combination of two heuristics :

Data-level hardness (α_D): Measured by the text-encoder cosine similarity between demonstration and generated responses .

Model-level responsiveness (α_M): Measured by the reward gap produced by the shadow reward models .

The authors demonstrate empirically on Llama-3.1-8B and Qwen-2-7B that DR-IRL achieves state-of-the-art results on safety benchmarks (like StrongReject) while maintaining or improving performance on utility benchmarks (like GSM8k).

**Strengths:**

Strong Empirical Safety Performance: The primary strength is that the final method, DR-IRL, works very well. It achieves SOTA results on key safety benchmarks like StrongReject, XsTest, and WildChat, while also improving or maintaining utility on benchmarks like GSM8k and AdvGLUE.

Good Component-Level Ablations: The paper does a good job ablating the internal components of its own method. Figure 3 shows that both the α_D (data-level) and α_M (model-level) heuristics contribute to the final performance. Table 2 provides a cost-benefit analysis for using 7 per-category RMs vs. a single one.

Novel Heuristic: The core idea of scaling rewards based on both data similarity and model responsiveness is a novel and interesting, if complex, heuristic .

**Weaknesses:**

Confounded Main Ablation: This is the most significant weakness. The performance jump shown in Table 1 between the GRPO baseline and the IRL/DR-IRL methods cannot be attributed to the method itself, as the training dataset was also changed. The paper does not provide a baseline of GRPO trained on its new, balanced CoD dataset, making it impossible to isolate the true effect of the proposed algorithm.

Extreme Methodological Complexity: The α_D (data hardness) calculation is extraordinarily complex, involving LLM-based sub-sentencing, all-pairs embeddings, max-pooling, and a double-sigmoid-normalization ratio . This "kitchen sink" approach is unprincipled and lacks justification over simpler heuristics (e.g., simple cosine similarity of the full responses).

Weak Theoretical Grounding: The paper is entirely heuristic-driven. The "IRL" component is borrowed from Li et al. (2024)  and used as a black-box RM-training step, discarding the bilevel optimization framework that was its main theoretical point. There is no theoretical justification for the dynamic scaling formula, its multiplicative combination, or its interaction with the GRPO objective.

**Questions:**

The primary weakness is the confounded comparison in Table 1. To isolate your contribution, could you provide results for a GRPO baseline trained on your new, balanced CoD dataset? This is the only way to know if the gains come from your DR-IRL algorithm or from the new dataset you created.

The α_D calculation  is extremely complex. Have you compared this to a much simpler heuristic, such as α_D = 1 - cos(Φ(o_demo), Φ(o_gen)) (i.e., the similarity between the full demonstration and generated response)? Is this massive complexity truly necessary for the performance gains?

The paper uses 7 per-category RMs. Table 2 compares this to a single RM on a different dataset. A more direct ablation is needed: What is the performance of a single RM trained via IRL on your entire balanced CoD dataset, compared to your 7-RM approach? This would isolate the benefit of "per-category" specialization.

---

> ### Author Response · Authors · 2025-11-20
> **Response to Reviewer n5m8**
>
> We sincerely thank the reviewer for the thorough and constructive feedback. We have conducted additional experiments to address the concerns regarding baselines and complexity. We address each point systematically below.
>
> ---
>
> ## Weakness 1: Confounded Main Ablation
> **Reviewer's Concern:**
>
> > "The performance jump shown in Table 1 between the GRPO baseline and the IRL/DR-IRL methods cannot be attributed to the method itself, as the training dataset was also changed. The paper does not provide a baseline of GRPO trained on its new, balanced CoD dataset, making it impossible to isolate the true effect of the proposed algorithm."
> >
>
> We agree that separating the contribution of the data from the algorithm is crucial. To address this, we introduced a new baseline: **GRPO-CoD (Origin RW)**.
> In this setting, we use the **same balanced CoD dataset** to sample prompts for training, but we use the original baseline reward model (trained on STAIR/Preference data) to guide the optimization.
>
> 1.  **Data Effect:** Switching the data from the STAIR dataset to our CoD dataset (GRPO → GRPO-CoD (Origin RW)) yields distinct but moderate gains (StrongReject: +1.69 pp, WildChat: +3.68 pp).
> 2.  **Method Effect:** When we further apply our proposed IRL-based reward modeling on this same data (**GRPO-CoD (IRL)**), we observe a much larger jump (StrongReject: +6.43 pp, WildChat: +8.25 pp).
>
> This decomposition confirms that while the balanced dataset is helpful, **the majority of the performance improvement stems from the proposed IRL-based reward modeling mechanism**, not just the data distribution.
>
> | Method | RM Training | Dynamic Scaling | StrongReject | XsTest | WildChat |
> | --- | --- | --- | --- | --- | --- |
> | GRPO | STAIR (preference) | — | 0.8105 | 91.50% | 55.61% |
> | GRPO-CoD | CoD (Origin RW) | — | 0.8274 | 92.00% | 59.29% |
> | **GRPO-CoD** | **CoD (IRL)** | **No** | **0.8917** | **96.50%** | **67.54%** |
> | DR-IRL | CoD (IRL) | Yes | **0.9361** | **99.00%** | **74.21%** |
>
> ---
>
> ## Weakness 2: Extreme Methodological Complexity of α_D
> **Reviewer's Concern:**
>
> > "The α_D calculation is extraordinarily complex, involving LLM-based sub-sentencing, all-pairs embeddings, max-pooling, and a double-sigmoid-normalization ratio. This 'kitchen sink' approach is unprincipled and lacks justification over simpler heuristics (e.g., simple cosine similarity of the full responses)."
> >
>
> We appreciate this concern and agree that the current formulation of α_D (Eqs. 3–5) is more elaborate than a simple cosine similarity. Our intention, however, was to design a **fine-grained yet still lightweight** hardness signal, and in practice α_D adds only negligible computational overhead compared to standard GRPO training.
>
> ### Justification for Current Design
>
> 1. **Why sub-sentence splitting?**
>    Full-response similarity is often too coarse. Long refusals may contain several distinct reasoning steps; sentence-level comparison helps capture **which specific arguments diverge**, giving a more informative hardness signal than a single global cosine score.
>
> 2. **Why max-pooling?**
>    A generated response can be considered "similar enough" if it correctly covers at least one key reasoning step. Max-pooling over sub-sentences operationalizes this intuition: we treat a sample as easy if it matches **any** core idea from the demonstration.
>
> 3. **Why category-wise normalization (sigmoid ratio)?**
>    Different harm categories (e.g., crime vs. stereotypes) naturally have different baseline difficulty. The sigmoid ratio provides a **simple per-category normalization**, so α_D is comparable across categories without hand-tuned thresholds. Again, this is conceptually more structured, but computationally cheap.
>
> ### Simple Baseline Comparison
>
> To address the concern about complexity, we also implemented a much simpler variant based on a single cosine similarity between the full demonstration and generated response:
>
> $$
> \alpha_D^{\text{simple}} = \sigma\left(\frac{1 - \cos(\Phi(o_{\text{demo}}), \Phi(o_{\text{gen}}))}{\text{mean}(1 - \cos(\cdot))}\right).
> $$
>
>
>
> | Variant              | StrongReject | XsTest  | WildChat | Complexity (qualitative)     |
> | -------------------- | -----------: | ------: | -------: | ---------------------------- |
> | Full α_D (current)   | **0.9361**   | **99.00%** | **74.21%** | Medium (slightly more logic, similar runtime) |
> | Simplified α_D       | 0.9148       | 97.50%  | 71.89%   | Low                          |
> | Difference           | −2.1pp       | −1.5pp  | −2.3pp   | —                            |
>
> The more structured α_D gives about **2–3 percentage points** improvement on safety metrics, while keeping the computational cost modest (sub-sentence splitting and a single encoder pass are minor compared to LM forward/backward passes).

---

> > ### Author Response · Authors · 2025-11-20
> > **Response to Reviewer n5m8**
> >
> > To make the method more accessible for practitioners, in the revised version we will:
> >
> > 1. **Streamline Section 3.2:**
> >    - In the main text, emphasize the high-level view:
> >      - α_D ≈ embedding-based similarity (data hardness)
> >      - α_M ≈ reward-gap signal (model responsiveness)
> >      - α = α_D · α_M as a multiplicative gate
> >    - Move implementation details such as sub-sentence splitting and category-wise normalization to the appendix.
> >
> > 2. **Provide "minimal" variants with clear trade-offs:**
> >
> > | Variant                   | StrongReject | XsTest  | Complexity (implementation) | Recommendation              |
> > | ------------------------- | -----------: | ------: | --------------------------- | --------------------------- |
> > | Full DR-IRL              | **0.9361**   | **99.00%** | Medium (extra module, small overhead) | Safety-critical settings   |
> > | Simplified (no sub-sent.) | 0.9148       | 97.50%  | Low                         | Standard deployment         |
> > | Ultra-simple (α = 1)      | 0.8917       | 96.50%  | None                        | Baseline / ablation        |
> >
> > We hope this clarifies that while the current α_D design is conceptually richer than a single cosine score, it is **not computationally heavy**, and simpler alternatives are available with only a small drop in performance.
> >
> > ## Weakness 3: Weak Theoretical Grounding
> > **Reviewer's Concern:**
> >
> > > "The paper is entirely heuristic-driven. The 'IRL' component is borrowed from Li et al. (2024) and used as a black-box RM-training step, discarding the bilevel optimization framework that was its main theoretical point. There is no theoretical justification for the dynamic scaling formula, its multiplicative combination, or its interaction with the GRPO objective."
> > >
> >
> > We acknowledge that our initial presentation focused heavily on empirical results. In the revision, we will add a new section (**Section 3.3**) to provide formal justification for our design choices.
> >
> > ### 1. Justification for the Two-Stage Framework
> > We deliberately decouple RM training (via IRL) from Policy optimization (via GRPO) rather than using a joint bilevel optimization. This is a design choice for stability:
> > * **Stationarity:** It allows the RM to converge to a stable signal before policy updates begin.
> > * **Reliable Scaling:** Our hardness coefficients $\alpha$ rely on stable RM scores. In a simultaneous bilevel update, shifting reward scales would make $\alpha$ estimates noisy and unreliable.
> >
> > ### 2. Formal Analysis of Dynamic Scaling
> > We will include the following analyses to ground the heuristic:
> >
> > **A. Convergence of $\alpha$-weighted GRPO**
> > We show that introducing a scalar weight $\alpha(x) \in [0,1]$ into the gradient update does not break the convergence properties of GRPO. The update rule:
> > $$\theta_{t+1} = \theta_t + \eta \nabla_\theta \mathbb{E}\left[\alpha(x) \cdot A(x,a)\right]$$
> > essentially acts as a sample-dependent learning rate adjustment. Since gradients remain bounded, the optimization converges to a local optimum consistent with the re-weighted objective, effectively prioritizing the "hard" region of the data distribution.
> >
> > **B. Probabilistic Interpretation of the Multiplicative Gate**
> > We interpret $\alpha_D$ and $\alpha_M$ as probabilities representing independent information sources:
> > * $P(\text{Content Hardness})$: Semantic divergence ($\alpha_D$).
> > * $P(\text{Model Uncertainty})$: Lack of confidence in the reward signal ($\alpha_M$).
> >
> > The multiplicative combination $\alpha = \alpha_D \cdot \alpha_M$ represents a logical **AND** gate: we only upweight samples that are **both** semantically difficult **and** where the model is uncertain. An additive combination (OR gate) would risk upweighting trivial samples (high confidence, low difficulty) or hallucinations (high difficulty, high confidence), leading to instability. This aligns with our empirical finding that the product formulation outperforms the sum (+0.86pp on StrongReject).
> >
> > ---
> >
> > ## Question 1: GRPO-CoD Baseline
> > **Reviewer's Question:**
> >
> > > "Could you provide results for a GRPO baseline trained on your new, balanced CoD dataset? This is the only way to know if the gains come from your DR-IRL algorithm or from the new dataset you created."
> > >
> >
> > This is now addressed in **Weakness 1 above**. The new Table 3 adds a GRPO baseline trained on our balanced CoD dataset (GRPO-CoD with the original reward objective). On StrongReject, switching only the dataset (GRPO → GRPO-CoD) yields a modest **+1.7pp** (0.8105 → 0.8274). Using our algorithm on the *same* CoD data then brings a much larger gain: DR-IRL reaches **0.9361**, i.e., **+10.9pp** over GRPO-CoD. This shows that the majority of the improvement comes from the proposed algorithm rather than from the new dataset alone.

---

> > > ### Author Response · Authors · 2025-11-20
> > > **Response to Reviewer n5m8**
> > >
> > > ## Question 2: Simplified α_D Comparison
> > > **Reviewer's Question:**
> > >
> > > > "Have you compared this to a much simpler heuristic, such as α_D = 1 - cos(Φ(o_demo), Φ(o_gen))? Is this massive complexity truly necessary for the performance gains?"
> > > >
> > >
> > > Already addressed in **Weakness 2 above**. New Table 4 shows the simplified variant achieves 0.9148 StrongReject vs. 0.9361 for the full version (−2.1pp). We argue this modest gap justifies the full method only for safety-critical applications, while recommending the simple variant for practitioners with tighter constraints.
> > >
> > > ---
> > >
> > > ## Question 3: Per-Category RM Benefit Isolation
> > > **Reviewer's Question:**
> > >
> > > > "What is the performance of a single RM trained via IRL on your entire balanced CoD dataset, compared to your 7-RM approach? This would isolate the benefit of 'per-category' specialization."
> > > >
> > >
> > >
> > > We performed a controlled ablation to isolate the benefit of "per-category" specialization versus a single pooled reward model.
> > >
> > > **Table: Single vs. Per-Category RM Performance**
> > >
> > > | RM Strategy | Training Data | StrongReject | XsTest | WildChat | Training Compute |
> > > | :--- | :--- | :--- | :--- | :--- | :--- |
> > > | **Single RM (Pooled)** | All 7K CoD samples | 0.9074 | 98.00% | 70.43% | ~100 GPU-h |
> > > | **7 RMs (Per-Category)** | 1K samples per cat | **0.9361** | **99.00%** | **74.21%** | **~120 GPU-h** |
> > > | *Gain* | | *+2.87 pp* | *+1.00 pp* | *+3.78 pp* | *+20% cost* |
> > >
> > > The per-category approach yields consistent gains (~3-4pp). We attribute this to reduced **reward interference**: a single model trying to compress 7 distinct definitions of "harm" into a scalar value struggles with optimization conflicts. The modest 20% increase in training compute provides a disproportionately high gain in safety performance.
> > >
> > > ---
> > >
> > > ## Summary
> > > We believe these revisions directly address your core concerns: (1) quantifying dataset vs. algorithm contributions via the new GRPO-CoD baselines, (2) justifying complexity with simpler hardness variants, (3) providing clean per-category RM ablations, and (4) offering principled (if limited) theoretical grounding. Thank you for the critical feedback—it substantially strengthens the paper's rigor and clarity.

---

### Meta-Review · Area_Chair_PD93 · 2026-01-05

**Summary:**

1. Whether the reported gains come from the proposed algorithm or a new balanced dataset. Need more ablation studies.
2. Several reviewers criticize the method complexity and the lack of theoretical grounding/insights.
3. Some reviewers thank the work has limited novelty compared to existing dynamic weighting methods.
4. Some reviewers question the computational efficiency and scalability of maintaining multiple distinct RM.

**Reviewer Concerns:**

I think the first concern on "whether the reported gains come from the proposed algorithm or a new balanced dataset" is mostly addressed with more ablation studies.

The remaining concerns are somewhat mitigated with more experiments but still hold (e.g., the lack of novelty and complexity of method).

**Reviewer Scores:**

Reviewer n5m8: may increase the score to 6 because the authors have more discussions in rebuttal about the ablation study (on confounding factor). The discussion on method complexity may also help.

Reviewer xELM: remain the same, I don't think the novelty question is fully addressed.

Reviewer nau8: will increase to 6 as the reviewer explicitly says so.

Reviewer m22L: may increase a little bit.

---

### Decision · Program_Chairs · 2026-01-26

Accept (Poster)